# Genome-Wide Identification and Characterization of U-Box Gene Family Members and Analysis of Their Expression Patterns in *Phaseolus vulgaris* L. under Cold Stress

**DOI:** 10.3390/ijms25147968

**Published:** 2024-07-21

**Authors:** Jiawei Wang, Zhiyuan Liu, Hongbing She, Zhaosheng Xu, Helong Zhang, Zhengwu Fang, Wei Qian

**Affiliations:** 1MARA Key Laboratory of Sustainable Crop Production in the Middle Reaches of the Yangtze River (Co-Construction by Ministry and Province), College of Agriculture, Yangtze University, Jingzhou 434025, China; jwwang9527@163.com; 2State Key Laboratory of Vegetable Biobreeding, Institute of Vegetables and Flowers, Chinese Academy of Agricultural Sciences, Beijing 100081, China; liuzhiyuan01@caas.cn (Z.L.); shehongbing@caas.cn (H.S.); xuzhaosheng@caas.cn (Z.X.); zhanghelong@caas.cn (H.Z.)

**Keywords:** *Phaseolus vulgaris* L., U-box genes, RNA-seq analysis, cold stress expression

## Abstract

The common bean (*Phaseolus vulgaris* L.) is an economically important food crop grown worldwide; however, its production is affected by various environmental stresses, including cold, heat, and drought stress. The plant U-box (PUB) protein family participates in various biological processes and stress responses, but the gene function and expression patterns of its members in the common bean remain unclear. Here, we systematically identified 63 U-box genes, including 8 tandem genes and 55 non-tandem genes, in the common bean. These *PvPUB* genes were unevenly distributed across 11 chromosomes, with chromosome 2 holding the most members of the PUB family, containing 10 PUB genes. The analysis of the phylogenetic tree classified the 63 PUB genes into three groups. Moreover, transcriptome analysis based on cold-tolerant and cold-sensitive varieties identified 4 differentially expressed *PvPUB* genes, suggesting their roles in cold tolerance. Taken together, this study serves as a valuable resource for exploring the functional aspects of the common bean U-box gene family and offers crucial theoretical support for the development of new cold-tolerant common bean varieties.

## 1. Introduction

During the growth and development process, plants are subjected to various abiotic stressors, such as low temperature, drought, and salinity [1]. These stressors can disrupt structural and functional expression in plant tissues, leading to oxidative damage and adverse effects on plant growth and development [2]. To cope with adverse environments, plants have evolved complex and sophisticated defense mechanisms. Previous studies have suggested that plants respond to abiotic stress primarily through multiple signaling pathways, encompassing transcriptional regulation [3], post-transcriptional modification [4], epigenetic regulation [5], and post-translational modification [6]. Multilayered gene expression regulation ensures that organisms can adapt precisely to environmental changes. Post-translational modifications, such as phosphorylation, glycosylation, and ubiquitination, significantly alter protein function [7], activity, stability, and cellular localization, ensuring proper protein function and dynamic regulation within the cell. These modifications are crucial for cellular signal transduction, metabolic regulation, and the response to environmental stress.

Ubiquitination is a post-translational modification process (PTM) in which small ubiquitin proteins are covalently attached to specific target proteins [8], regulating diverse cellular processes in higher plants and vertebrates, including cell death, cell division, hormone responses, and biotic and abiotic stress responses [9]. Ubiquitin, which comprises 76 amino acids, exists in free form and can be conjugated to proteins either individually (mono-ubiquitination) or in multiple units (poly-ubiquitination) [10]. Within the ubiquitin–proteasome system (UPS), ubiquitin is linked to substrates in the form of Lys-48 or Lys-11-linked polyubiquitin chains, serving as substrate degradation signals [11]. A recent study has indicated that the UPS is involved in biotic and abiotic stress responses, hormone regulation, and various plant development pathways [12].

The U-box gene family encompasses genes encoding the U-box domain, which was first identified in yeast as part of the Ubiquitin Fusion Degradation 2 (UFD2) protein [13]. Proteins encoded by these genes typically exhibit ubiquitin ligase (E3) activity, playing a pivotal role in the regulation of protein ubiquitination modifications. During protein degradation via ubiquitination, ubiquitin ligase (E3) is involved in a cascade of enzymatic reactions, facilitated by the E1 ubiquitin-activating enzyme, the E2 ubiquitin-conjugating enzyme [14], and the E3 ubiquitin ligase [15]. During this process, the size and composition of E3 ligase play a pivotal role in substrate specificity, which is crucial for recognizing different substrates. Based on their recognition mechanisms and subunit composition, E3 ubiquitin ligases are primarily classified into four main types: U-box, Homology to E6-Associated Carboxy-Terminus (HECT) [16], Really Interesting New Gene (RING) [17], and Cullin-RING ligases (CRLs) [18].

Previous studies have shown that E3 ubiquitin ligases with HECT or RING domains have unique characteristics [15]. The HECT domain forms a thioester bond with ubiquitin, directly transferring it to the substrate. The RING domain facilitates the transfer of ubiquitin from the E2 enzyme to the substrate [10]. U-box genes contain Armadillo (ARM) repeats, which enhance substrate interactions, recognize various eukaryotic proteins, and control multiple biological functions [19]. Ubiquitination plays diverse roles, including protein degradation, protein–protein interactions, subcellular localization, and kinase activation.

U-box (PUB) genes are primarily found in plants. For example, *Arabidopsis thaliana* has 61, pear has 63 [20], cabbage has 99, and soybean has 125 PUB genes. Studies have shown the involvement of PUB proteins in various biological processes and stress responses. A total of 63 *MdPUB* genes have been found in apples, with *MdPUB23* regulating cold stress by degrading *MdICE1* [21]. In addition to the U-box, the WRKY and bHLH gene families play significant roles in plant stress responses. *lbbHLH33* enhances cold resistance [22], while bHLH factors are involved in anthocyanin biosynthesis, flower development, and stress responses [23]. Thus, the importance of the U-box gene family in plants is apparent.

Low temperatures significantly impact crop growth and yield worldwide, posing a major threat to agriculture by limiting vegetation distribution and severely hampering agricultural development. Consequently, addressing the impact of low-temperature stress on crops has become an urgent and essential issue, and analyzing the molecular mechanisms underlying plant responses to low-temperature stress is imperative. Previous studies on low-temperature stress encompass signal perception, signal transduction, functional gene expression, and a range of physiological and cellular mechanisms induced by low temperatures [24]. E3 ubiquitin ligases play a pivotal role in this process. They enhance plant cold tolerance by modulating signal transduction pathways. Under favorable growth conditions, they suppress the transmission of low-temperature stress signals and serve as positive feedback factors to enhance signal transduction [25].

The plant U-box family has been extensively studied for its role in abiotic stress responses, especially in *Arabidopsis*, apples, and rice. However, few studies have evaluated this family in legumes. The common bean (*Phaseolus vulgaris* L.), a significant economic vegetable crop worldwide, is classified into dry beans and snap beans based on their edible parts. Snap beans are rich in fiber, protein, vitamins, and minerals. However, abiotic stress hinders the growth and development and reduces the yield of common beans. Therefore, understanding the genetic factors behind the common bean’s tolerance to cold, drought, and salinity stress is crucial for agricultural development. In this study, we utilized the common bean genome (Pvulgaris_442_v2.1) to systematically analyze the 63 identified *PvPUB* genes. RNA sequencing (RNA-seq) identified differentially expressed genes related to cold stress, offering new insights into the molecular mechanisms of cold tolerance in *P*. *vulgaris*.

## 2. Results

### 2.1. Identification and Characterization of the U-Box Gene Family in Phaseolus vulgaris

A comprehensive search of the common bean genome (Pvulgaris_442_v2.1) was conducted using the HMMER (Hidden Markov Model) search tool, and 63 U-box genes were identified. These genes all contained U-box domains, some of which were part of composite domains. Each of these 63 genes was individually confirmed to contain a complete U-box domain through the InterPro online tool (https://www.ebi.ac.uk/interpro/search/text/, accessed on 23 April 2024). These genes were numbered based on their genomic positions with the prefix *PvPUB* (Table 1).

The molecular weights of these *PvPUB* genes ranged from 31.65 to 167.07 kDa, indicating significant diversity in protein size and potential biological functions. The amino acid lengths of these proteins ranged from 278 to 1013, further highlighting the diversity in their structural complexity and functional potential. This range suggests that these proteins may have varied roles, with longer proteins potentially containing multiple functional domains and participating in more complex biological processes. The isoelectric points (Isoelectric Point) ranged from 4.72 to 9.31, with an average of 6.85, suggesting varied stability and functionality in different cellular environments. The GRAVY values ranged from −0.7 to 0.2, indicating that most of these proteins were hydrophilic and suited to aqueous environments.

### 2.2. Chromosomal Localization of the U-Box Gene Family in Phaseolus vulgaris

We identified 63 *PvPUB* genes distributed across the chromosomes of the common bean. The distribution of genes among chromosomes varied significantly (Figure 1). Chromosome 2 contained the most genes, with a total of 10, followed by chromosome 8, with 9 genes. Chromosomes 7 and 9 each contained 7 genes, and chromosome 3 had 6 genes. Chromosomes 6, 10, and 11 each had 4 genes, while chromosomes 1, 4, and 5 each had 3 genes. In addition, three genes were located on unassembled contigs.

Interestingly, the distribution of U-box genes was not correlated with chromosome length. Chromosome 6, which was the shortest chromosome, contained four genes, while chromosome 11, which was the longest chromosome, also contained four genes (Figure 1). This indicates the uneven distribution of U-box genes across chromosomes. Furthermore, four pairs of tandem duplicated genes, which may play a crucial role in the expansion and functional diversity of the *PvPUB* gene family, were identified.

### 2.3. Phylogenetic Analysis and Classification of the U-Box Gene Family in Phaseolus vulgaris

To infer the evolutionary relationships and corresponding functions of U-box genes in common bean, we constructed a phylogenetic tree based on the amino acid sequences of the 63 identified *PvPUB* genes using MEGA-11 software(v11.0.13). Based on this analysis, the *PvPUB* genes were classified into distinct groups, as shown in Figure 2.

The phylogenetic tree revealed three major classes: Class 1 included 16 genes containing only the U-box domain; Class 2 comprised 34 genes containing the RING-UBOX_PUB domain; and Class 3 consisted of 13 genes containing various domains, such as RING-BOX_(UBEA4, CHIP, WDSUB1).

To elucidate the evolutionary relationships and functional implications of PUB genes in common bean, we constructed a phylogenetic tree using the amino acid sequences of 61 *AtPUB* genes from *A*. *thaliana* [26] and the protein sequences of *PvPUB* genes from common bean (Figure 3). The phylogenetic analysis grouped 124 genes into 8 subgroups: S1, S2, S3, S4, S5, S6, S7, and S8. Subgroup S3 contained the most genes (33), and S7 had the least (7). Both subgroups S2 and S5 each contained 8 genes. Subgroup S1 mainly contained Class 3 genes, excluding *PvPUB47*. Subgroups S2, S3, S5, S6, S7, and S8 contained 2, 9, 1, 1, 1, and 2 Class 1 genes, respectively, and 3, 7, 3, 5, 2, and 5 Class 2 genes, respectively. By contrast, subgroup S4 contained 4 Class 2 genes and 1 Class 3 gene. A similar number of PUB genes was observed in both species, suggesting a conserved gene family size and indicating essential and possibly conserved functions. In subgroup S2, there is a separate branch that only includes common bean PUB genes (*PvPUB16*, *PvPUB56*, and *PvPUB48*). This suggests that these genes may have unique functions in the common bean. Similarly, in subgroup S4, there is a separate branch that only includes Arabidopsis PUB genes (*AtPUB49*, *AtPUB60*, and *AtPUB59*), indicating that these genes may have unique functions in *Arabidopsis*. This implies that the functions of these genes in Arabidopsis may not have corresponding gene expressions in the common bean (highlighted in red in Figure 3).

### 2.4. Analysis of Conserved Motifs, Domains, and Structure of U-Box Genes in Phaseolus vulgaris

To investigate the structural evolution of the *PvPUB* genes, we analyzed their conserved motifs, domains, and gene structures in conjunction with their phylogenetic grouping (Figure 4). A total of 20 motifs were used as the identification standard (Figure 4A). Motifs 1 and 2 were universally present in all *PvPUB* genes, and motifs 3–6 were frequently observed.

Gene structure analysis revealed significant diversity in the number of exons among the *PvPUB* genes (Figure 4B). A total of 25 genes contained a single exon, and 10 genes contained four exons. Subsequent analysis of conserved domains highlighted the structural characteristics of *PvPUB* genes. The *PvPUB* gene family included at least two core conserved domains (U-box and RING-UBOX_PUB) and several specific RING-UBOX domains (UBE4A, CHIP, WDSUB1, ARM, and HEAT) (Figure 4C).

### 2.5. Subcellular Localization and Cis-Acting Element Analysis of U-Box Genes in Phaseolus vulgaris

To elucidate the functional roles of the *PvPUB* gene family in common bean, we predicted the subcellular localization of the 63 identified *PvPUB* genes using the WoLF PSORT website. Collectively, these data provide a foundation for further functional studies of U-box genes in *P*. *vulgaris*. Based on subcellular localization, 19, 18, and 16 genes are localized in the chloroplast, nucleus, and cytoplasm, respectively. Two genes each localized in the endoplasmic reticulum (*PvPUB40*, *PvPUB58*) and Golgi apparatus (*PvPUB38*, *PvPUB48*), and one gene each localized in the mitochondria (*PvPUB2*) and peroxisome (*PvPUB45*). Four genes are localized in the Plasma membrane (*PvPUB9*, *PvPUB17*, *PvPUB44*, and *PvPUB46*). As shown in Figure 5, different distribution patterns of cis-acting elements were observed in the promoter regions of the *PvPUB* genes.

The promoter regions of the *PvPUB* genes in common bean were enriched with diverse cis-acting elements, including elements responsible for light response, hormone response (abscisic acid, salicylic acid, gibberellin, methyl jasmonate, auxin, and flavonoid biosynthesis), low-temperature response, cell cycle regulation, and defense and stress responses. Light-responsive elements were the most prevalent, followed by hormone-responsive elements and those involved in low-temperature and drought stress responses. Our analysis revealed several key findings regarding the promoter regions of *PvPUB* genes in common bean. MYB binding sites associated with flavonoid biosynthesis were identified in *PvPUB39*, *PvPUB44*, *PvPUB22*, *PvPUB48*, and *PvPUB59*. Cis-acting elements related to low-temperature response were predicted in 28 *PvPUB* genes. MYB binding sites associated with drought response were found in 29 *PvPUB* genes. These results indicate that *PvPUB* genes are significantly involved in flavonoid biosynthesis and responses to low-temperature and drought stress, underscoring their crucial roles in plant physiology and stress adaptation.

### 2.6. Tandem Duplication and Collinearity Analysis of U-Box Genes in Phaseolus vulgaris

By studying the *PvPUB* gene family in common bean, we identified four pairs of tandemly duplicated genes, as follows: Phvul.005G119100.1.v2.1 (*PvPUB24*) and Phvul.005G119200.1.v2.1 (*PvPUB25*); Phvul.008G13440.1.v2.1 (*PvPUB41*) and Phvul.008G134500.1.v2.1 (*PvPUB42*); Phvul.007G267700.1.v2.1 (*PvPUB35*) and Phvul.007G267800.1.v2.1 (*PvPUB36*); Phvul.011G099300.1.v2.1 (*PvPUB59*) and Phvul.011G099400.1.v2.1 (*PvPUB60*).

Moreover, intra-species collinearity analysis identified 20 pairs of homologous genes, encompassing 33 homologous genes, indicating a significant degree of gene duplication and conservation within the *PvPUB* gene family, which may contribute to the functional diversification and evolutionary adaptation of these genes in common bean (Figure 6).

To elucidate the evolutionary relationships of the *PvPUB* gene family with other species, we performed a collinearity analysis between the common bean and three other species: *A*. *thaliana*, *Medicago truncatula*, and *Oryza sativa* (Figure 7).

A total of 32 *PvPUB* genes in the common bean had collinear counterparts in *Arabidopsis*, resulting in 51 collinear gene pairs. Specifically, *PvPUB3*, *PvPUB11*, *PvPUB35*, and *PvPUB50* each had three collinear gene pairs in *A*. *thaliana*. *PvPUB17* and nine other *PvPUB* genes had two collinear gene pairs each. The remaining 18 *PvPUB* genes each had a single collinear gene pair. In the collinearity analysis, 48 *PvPUB* genes in common bean corresponded to 76 collinear gene pairs in *M*. *truncatula*. Among these, 6 *PvPUB* genes had 3 collinear gene pairs each, and 16 genes had 2 collinear gene pairs each. The remaining genes were single collinear gene pairs.

A total of 18 *PvPUB* genes in common bean corresponded to 30 collinear gene pairs in *O*. *sativa*. *PvPUB1* had four collinear gene pairs, while *PvPUB24* and *PvPUB35* each had three collinear gene pairs. Five other *PvPUB* genes each had two collinear gene pairs, and the remaining genes were single collinear gene pairs. Interestingly, across these species, 16 *PvPUB* genes in common bean were consistently involved in collinear relationships and were highly conserved, indicating that they perform similar functions across the four species.

### 2.7. Expression Patterns of PvPUB Genes under Low-Temperature Stress

We aimed to identify the *PvPUB* genes associated with low-temperature stress. We utilized transcriptome data from the NCBI public database, specifically from BioProject PRJNA793687 [27]. Experimental conditions: T was the treatment group (4 °C for 3 days), and C was the control group (25 °C for 3 days). Our analysis revealed differential expression of U-box genes under low-temperature stress between two common bean varieties. In cold-sensitive variety 093, a total of 13 genes were upregulated and seven were downregulated. In the cold-tolerant variety 120, a total of 16 genes were upregulated and six were downregulated. According to the expression heatmap (Figure 8), eight *PvPUB* genes, including *PvPUB45*, *PvPUB57*, *PvPUB23*, *PvPUB2*, *PvPUB46*, *PvPUB8*, *PvPUB3,* and *PvPUB53*, showed higher expression levels in cold-sensitive variety 093. By contrast, 14 *PvPUB* genes, including *PvPUB17*, *PvPUB5*, *PvPUB44*, and *PvPUB29*, exhibited higher expression levels in the cold-tolerant variety 120. Genes highly expressed in variety 093 tended to be downregulated in variety 120, suggesting a potential role in negative regulation. This differential expression pattern indicates that these *PvPUB* genes play crucial roles in the response to low-temperature stress, potentially contributing to cold tolerance in these common bean varieties.

## 3. Discussion

We identified 63 *PvPUB* genes unevenly distributed across 11 chromosomes and 3 unassembled scaffolds, with 5 pairs of duplicated genes indicating a complex evolutionary history involving segmental duplication and chromosomal rearrangement. These gene duplications likely expand functional diversity, aiding adaptation to various environmental challenges. Our analysis of cis-acting elements revealed that *PvPUB* genes are crucial in abiotic stress responses and hormone signaling, particularly under extreme conditions that affect plant growth, development, and crop yield. Plants face severe impacts under adverse temperatures, necessitating adaptations in their growth processes, such as thermoperiodism, thermomorphogenesis, vernalization, cold acclimation, and responses to extreme temperatures [28]. Specifically, chilling injury at 0–15 °C [29] causes significant reductions in photosynthetic efficiency and cellular damage due to impaired photosynthetic enzymes, reduced photosystem activity, and phase transitions in membrane lipids that increase rigidity and permeability. These physiological changes underscore the vital roles of *PvPUB* genes in enhancing plant resilience to low-temperature stress, which is critical for maintaining plant health and productivity.

In response to drought stress, the expression of the *AtPUB22* and *AtPUB23* genes in *A*. *thaliana* is significantly upregulated [30]. Overexpression of these genes enhances drought tolerance by modulating the abscisic acid (ABA) signaling pathway and reactive oxygen species (ROS) metabolism [31]. In this study, homologous genes *PvPUB1* and *PvPUB24* showed increased expression under cold stress. Given that both drought and cold stress can induce similar physiological responses, such as ABA accumulation and ROS production [6], *PvPUB1* and *PvPUB24* may play a regulatory role in drought stress.

Regarding low-temperature stress, several PUB genes play crucial roles in enhancing cold tolerance through various mechanisms. In *Arabidopsis*, *AtPUB18* and *AtPUB19* regulate the stability of low-temperature responsive transcription factors via ubiquitination [32], thereby improving cold tolerance. Similarly, homologous genes *PvPUB5* and *PvPUB17* exhibited increased expression under cold stress, suggesting that they share similar functions in enhancing plant cold tolerance. In addition, a novel U-box type E3 ubiquitin ligase, *VaPUB*, has been isolated from the cold-hardy grapevine ‘Zuoshanyi’ (*Vitis amurensis* Rupr. cv.). *VaPUB* overexpression in *Arabidopsis* significantly enhanced cold and salt stress tolerance. *VaPUB* regulated the cold stress response by upregulating *CBF1/DREB1B*, *CBF3/DREB1A*, and several cold-inducible genes and repressing *CBF2/DREB1C* [33]. This further supports the critical role of PUB genes in stress response regulation.

Ubiquitination is a critical post-translational modification in which ubiquitin molecules are covalently attached to target proteins [34]. This process involves a cascade of enzymatic reactions coordinated by the E1 activating enzyme, the E2 conjugating enzyme, and the E3 ligase, impacting protein stability, activity, and interactions [35]. Ubiquitination regulates numerous physiological processes, such as cell proliferation, apoptosis, autophagy, and signal transduction, and is essential for immune responses [36,37]. Under stress conditions, E3 ubiquitin ligases mark proteins for degradation using the 26S proteasome, a key adaptation mechanism to environmental stress [38].

Previous studies have demonstrated the crucial roles of ubiquitin ligases in mediating plant responses to various abiotic stressors. For example, ubiquitin ligase SDIR1 mediates the interaction between abscisic acid (ABA) and jasmonic acid (JA) signaling pathways in *Arabidopsis*, enhancing the plant’s resilience to drought and salt stress [39]. Another study revealed the importance of a RING-finger E3 ligase in wild tomato species that significantly increases salt tolerance by stabilizing proteins through ubiquitination [40]. Specific RING-UBOX_PUB domain-containing E3 ligases in *Arabidopsis* negatively regulate the DREB2A transcription factor, effectively balancing drought stress responses with growth [41]. In addition, studies on tomatoes have shown that ubiquitination plays a significant role in cold tolerance by regulating the degradation of key proteins through the RING-UBOX_PUB domain [42].

Analysis of publicly available transcriptome data of the common bean under cold stress revealed differential expression levels for *PvPUB* genes. Four *PvPUB* genes—*PvPUB17*, *PvPUB5*, *PvPUB44*, and *PvPUB29*—showed enhanced expression in the cold-tolerant bean variety. These genes contain the RING-UBOX_PUB domain, which is typically found in E3 ubiquitin ligases [43]. This domain comprises several key components: the RING-finger, which mediates the interaction between the E3 ubiquitin ligase and the E2 ubiquitin-conjugating enzyme to facilitate ubiquitin transfer [44], the U-box, which stabilizes the E3-E2 enzyme complex, and the PUB domain, which binds ubiquitin molecules. Additionally, *PvPUB17*, *PvPUB5*, *PvPUB44*, and *PvPUB29* possess cis-acting elements involved in low-temperature responsiveness, abscisic acid responsiveness, and MeJA-responsiveness. These elements play important roles in plant responses to various abiotic stresses, including low temperatures.

The identification of *PvPUB* genes and their domain structures offers insights into their roles in cold stress responses. The presence of the RING-UBOX_PUB domain indicates that these genes function as E3 ubiquitin ligases, similar to SDIR1 in *Arabidopsis*, which mediates ABA and JA signaling, and SpRing, which enhances salt tolerance. The involvement of these domains in protein ubiquitination underscores their role in regulating protein stability and stress responses. These findings are consistent with previous studies highlighting the critical roles of E3 ligases in stress tolerance. Therefore, the identified *PvPUB* genes may enhance cold tolerance in common beans. Further characterization of these genes may facilitate the development of cold-resistant bean varieties, thereby contributing to agricultural sustainability and food security.

## 4. Materials and Methods

### 4.1. Identification of the U-Box Gene Family in Phaseolus vulgaris

To identify U-box family members in common bean, we utilized the HMMMER search for the U-box domain (PF04564) from the Pfam database (http://pfam.sanger.ac.uk/, accessed on 23 April 2024). This was used as a query to perform an HMMMER search within the common bean genome with an E-value cut-off of 1 × 10^−5^. Through the InterPro online tool, each of the 63 identified genes was individually confirmed to contain a complete U-box domain (Search—InterPro https://www.ebi.ac.uk/interpro/search/text/, accessed on 23 April 2024). For further identification, we searched for the U-box domain of all sequences to verify whether they belonged to the U-box gene family and classified them by U-box domain.

### 4.2. Phylogenetic Analysis and Classification of the U-Box Gene Family in Phaseolus vulgaris

We conducted a two-part analysis. In the first part, we analyzed the 63 identified *PvPUB* genes, grouping them into Classes 1–3 based on the presence of specific domains, such as U-box, RING-finger_PUB, and RING-BOX (UBEA4, CHIP, WDSUB1). In the second part, we performed a phylogenetic analysis comparing the common bean and *Arabidopsis* U-box genes. We used the protein sequences of 61 Arabidopsis U-box family genes(GeneFamily—pub—TAIR https://www.arabidopsis.org/browse/gene_family/pub, accessed on 25 April 2024) and the identified *PvPUB* genes. Multiple sequence alignment was conducted using the MUSCLE method in MEGA11 (https://www.megasoftware.net/, accessed on 27 April 2024), and a phylogenetic tree was constructed using the neighbor-joining method. We visualized the phylogenetic relationships between common bean and *Arabidopsis* U-box genes using the TVBOT tool in Chiplot (https://www.chiplot.online/tvbot.html, accessed on 28 April 2024).

### 4.3. Analysis of Conserved Motif, Domain, and Gene Structure of the U-Box Gene in Phaseolus vulgaris

We used the online tool MEME (MEME—Submission form https://meme-suite.org/meme/tools/meme, accessed on 28 April 2024) to analyze conserved motifs [45]. The PvPUB protein sequences were subjected to NCBI’s conserved domain search (Welcome to NCBI Batch CD-search https://www.ncbi.nlm.nih.gov/Structure/bwrpsb/bwrpsb.cgi, accessed on 29 April 2024) to confirm the presence of U-box domains. Using the GFF3 file of the common bean genome (Pvulgaris_442_v2.1), we extracted CDS and UTR region information with TBtools II (About TBtools—CJchen’s Blog, https://cj-chen.github.io/tbtools/, accessed on 30 April 2024). The gene structure, motifs, and domain data obtained from these analyses were then visualized using TBtools II.

### 4.4. Subcellular Localization and Cis-Acting Element Analysis of the U-Box Gene in Phaseolus vulgaris

We predicted the subcellular localization of 63 *PvPUB* genes in common bean using the WoLF PSORT tool (Protein Subcellular Localization Prediction https://wolfpsort.hgc.jp/, accessed on 1 May 2024). To identify cis-acting regulatory elements within 2000 bp upstream of these genes, we used the PlantCARE database (PlantCARE, a database of plant promoters and their cis-acting regulatory elements https://bioinformatics.psb.ugent.be/webtools/plantcare/html/, accessed on 3 May 2024) [46]. The positions, numbers, and types of cis-acting elements were visualized using TBtools II.

### 4.5. Chromosome Localization and Collinearity Analysis of the U-Box Gene in Phaseolus vulgaris

The length, start, and end positions of the *PvPUB* genes were obtained from the common bean genome (Pvulgaris_442_v2.1) and its GFF3 file. These data were visualized using the Gene Location Visualize feature in TBtools II. Intraspecies collinearity of common bean U-box genes was analyzed and visualized using TBtools II.

For interspecies collinearity analysis, we downloaded the genomes and GFF3 files of *Arabidopsis*, alfalfa, and rice from LegumeIP V3 (LegumeIP V3: From Models to Crops—An Integrative Gene Discovery Platform for Translational Genomics in Legumes (LegumeIP V3: From Models to Crops—An Integrative Gene Discovery Platform for Translational Genomics in Legumes (zhaolab.org)), accessed on 5 May 2024). The results were visualized using TBtools II.

### 4.6. RNA-Seq Analysis

The RNA-seq data used in this study were obtained from a previous study [27]. RNA was extracted from the true leaves of two common bean varieties: 093 (cold-sensitive) and 120 (cold-tolerant). The transcriptome data were stored in the NCBI database under BioProject ID: *PRJNA793687* (Integrated transcriptomics and metabolomics analysis (https://www.ncbi.nlm.nih.gov/bioproject/793687), accessed on 10 May 2024).

We first filtered the raw RNA-seq reads using Fastp (v0.20.0) [47]. The clean reads were then aligned to the common bean reference genome (Pvulgaris_442_v2.1) using HISAT2 (v4.8.2) with default parameters [48]. Read counts for each gene were estimated using featureCounts (v2.0.1) [49] and converted to transcripts per million (TPM). TPM values were normalized using R (v4.3.1) (https://www.r-project.org/, accessed on 15 May 2024) and visualized with Chiplot (https://www.chiplot.online/tvbot.html, accessed on 20 May 2024). Differentially expressed genes between the cold-sensitive and cold-tolerant varieties (both treated at 4 °C for 3 days) were identified using the DESeq2 package in R, with the following criteria: *p* < 0.05 and fold change (FC) > 2.

## 5. Conclusions

In this study, we first identified 63 U-box gene family members in common bean. We then constructed a phylogenetic tree using the neighbor-joining method, combining the amino acid sequences of these *PvPUB* genes with those of *Arabidopsis* U-box proteins to analyze their evolutionary relationships. Subsequently, we examined the conserved motifs, domains, and gene structures of these sequences, revealing that all *PvPUB* genes contained highly conserved U-box domains, with some existing as composite domains. Cis-acting regulatory element analysis showed that these genes may be involved in auxin, gibberellin, and abscisic acid synthesis, as well as circadian rhythm, cell differentiation, and cold response. Tandem duplication and collinearity contributed to the functional diversification and evolution of the common bean genome. Transcriptome data of different bean varieties under identical treatment conditions suggests that *PvPUB17*, *PvPUB5*, *PvPUB44*, and *PvPUB29* are involved in the cold stress response. These insights enhance our understanding of the U-box gene family’s potential roles in the common bean, providing valuable targets for improving cold stress tolerance in crops.

## Figures and Tables

**Figure 1 ijms-25-07968-f001:**
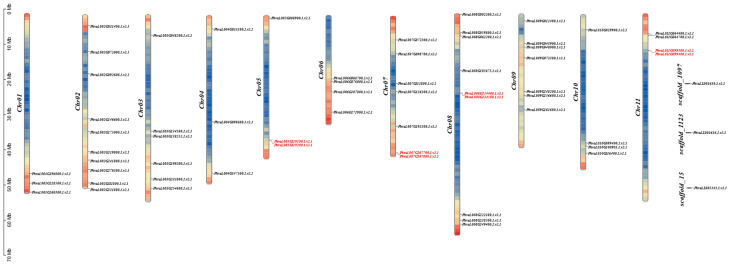
The 63 U-box genes are distributed across the 11 chromosomes and 3 unassembled scaffolds in the common bean genome (Pvulgaris_442_v2.1). Tandem duplications are highlighted in red, and chromosomes are color-coded by gene density (blue to red gradient). Visualization was performed using TBtools II (https://cj-chen.github.io/tbtools/, accessed on 30 April 2024).

**Figure 2 ijms-25-07968-f002:**
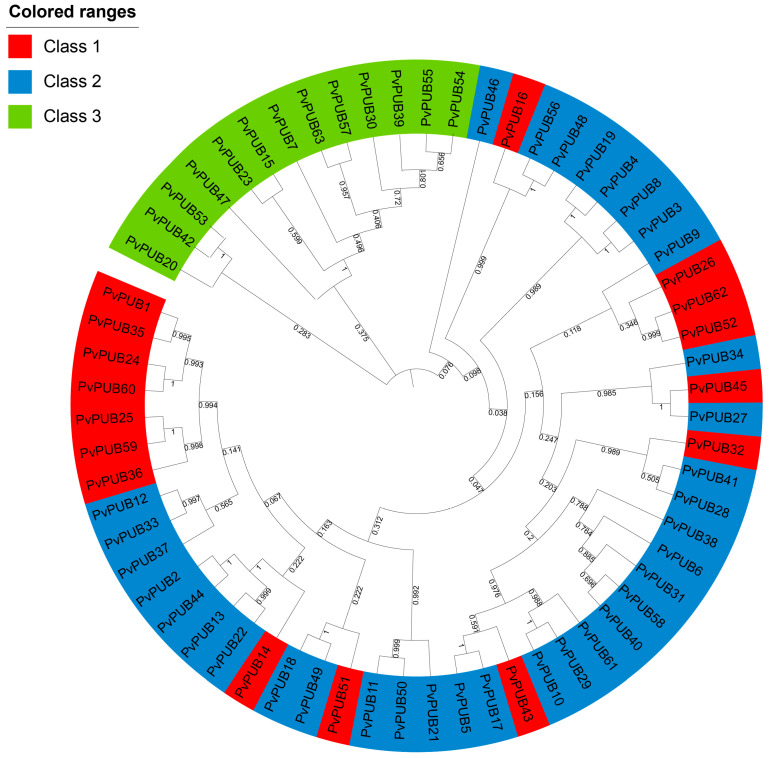
A phylogenetic tree of the common bean U-box gene family members was constructed, revealing three distinct classes, each represented by a different color. Protein multiple sequence alignment and visualization were performed using MEGA11 (v11.0.13), with modifications made using the Multiple Alignment Trimming feature in TBtools II. Data visualization and enhancement were conducted using TVBOT.

**Figure 3 ijms-25-07968-f003:**
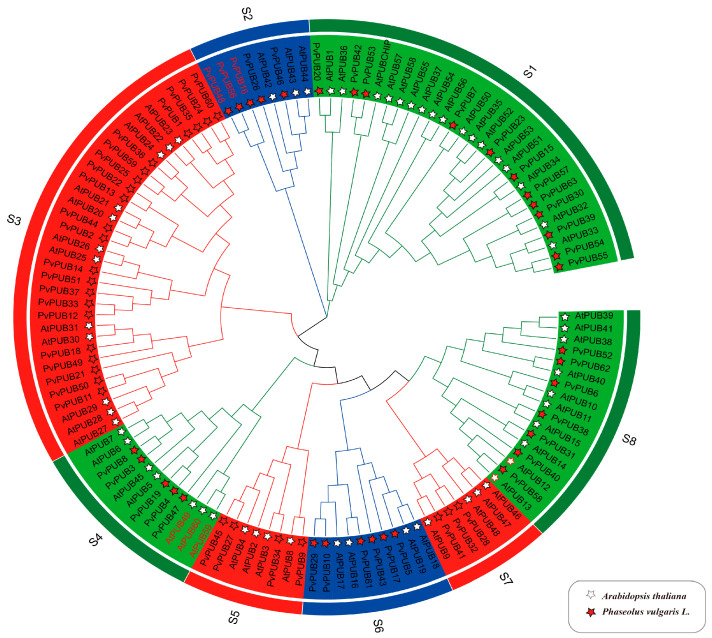
The phylogenetic tree of PUB genes from common bean and *Arabidopsis* was constructed using amino acid sequences. Common bean genes are marked with red pentagrams, and *Arabidopsis* genes are marked with white pentagrams. Different colored lines in the outermost ring denote distinct subgroups. Protein multiple sequence alignment and visualization were performed using MEGA11 (v11.0.13), refined in R, and visualized using Chiplot (https://www.chiplot.online/tvbot.html, accessed on 28 April 2024).

**Figure 4 ijms-25-07968-f004:**
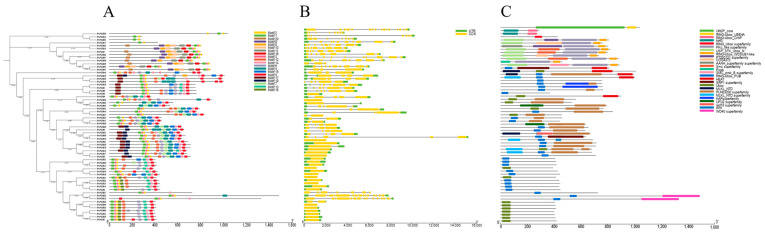
Conserved motif (**A**), gene structure (**B**), and conserved domain (**C**) analysis of common bean *PvPUB* genes. Different colored blocks represent distinct conserved motifs (**A**), and each is labeled with corresponding numbers. In the conserved domain (**C**) analysis, various colors indicate different domains. In the *PvPUB* gene structure (**B**), green boxes denote UTRs, yellow boxes represent CDSs, and horizontal lines within the boxes indicate introns. Motifs (**A**) were identified using the MEME online tool, and conserved domains (**C**) were determined via the NCBI CD search. The final visualization was performed using TBtools II.

**Figure 5 ijms-25-07968-f005:**
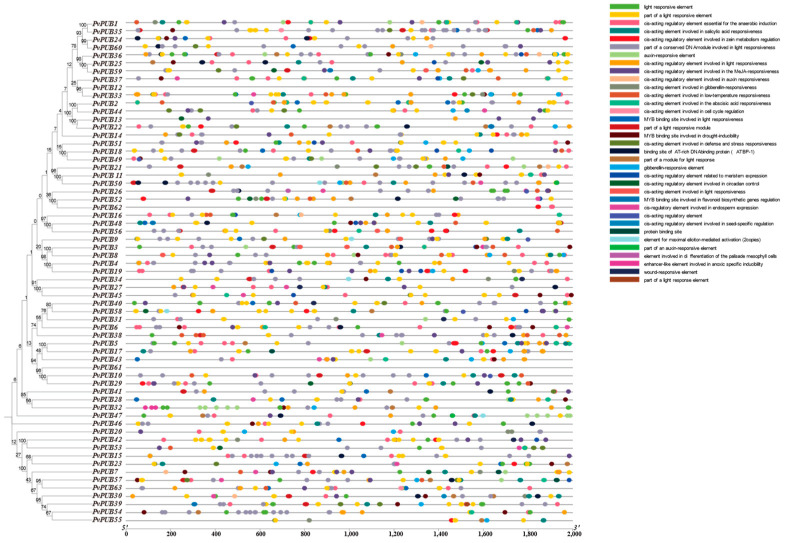
Cis-acting element analysis of the 63 predicted *PvPUB* genes. The distribution of 35 cis-acting elements in the promoters of the common bean PUB gene family is illustrated alongside the phylogenetic tree. Colored circles on the black lines represent different cis-acting elements. Cis-acting elements were predicted using the online tool PlantCARE and visualized with TBtools II.

**Figure 6 ijms-25-07968-f006:**
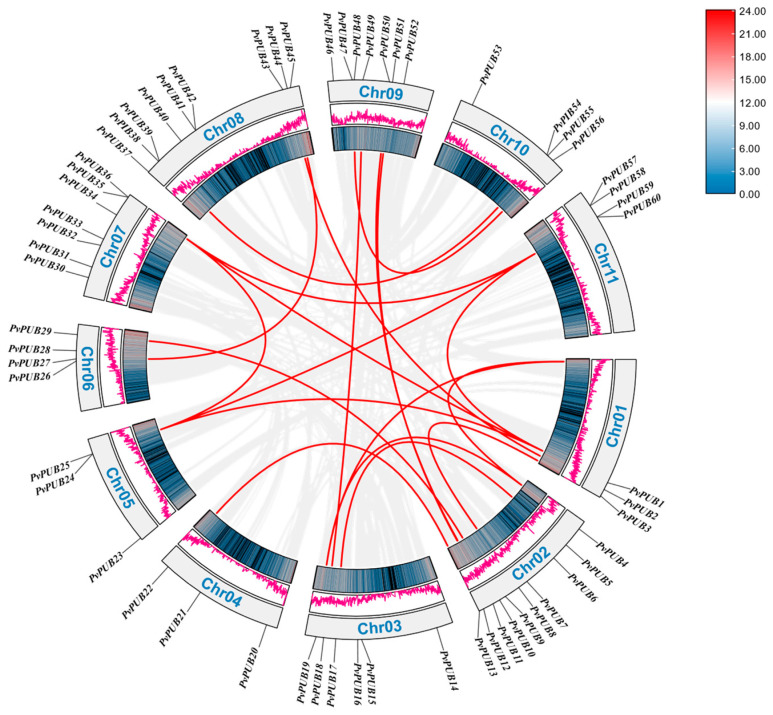
Intraspecies collinearity within the common bean genome. Black gene IDs represent *PvPUB* genes, while red lines indicate collinear gene pairs across different chromosomes. The inner blue gradient circles and pink curves denote the gene density on the respective chromosomes. Data visualization was performed using TBtools II.

**Figure 7 ijms-25-07968-f007:**
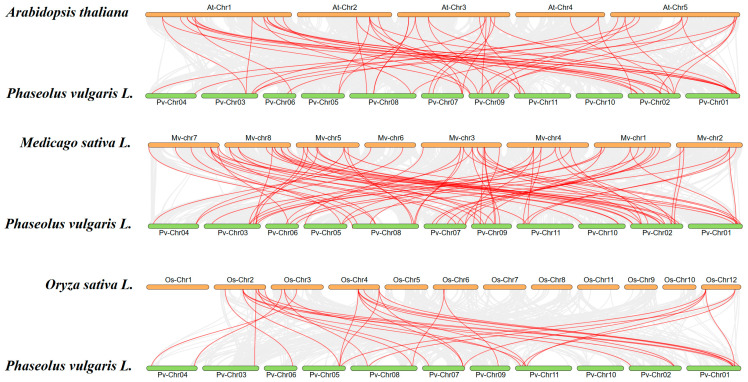
Genomic collinearity analysis among common bean, *Arabidopsis thaliana* (At), *Medicago sativa* L. (Mv), and *Oryza sativa* L. (Os). Gray lines connect collinear gene pairs, and red lines highlight collinear pairs involving *PvPUB* genes. Visualization was performed using TBtools II.

**Figure 8 ijms-25-07968-f008:**
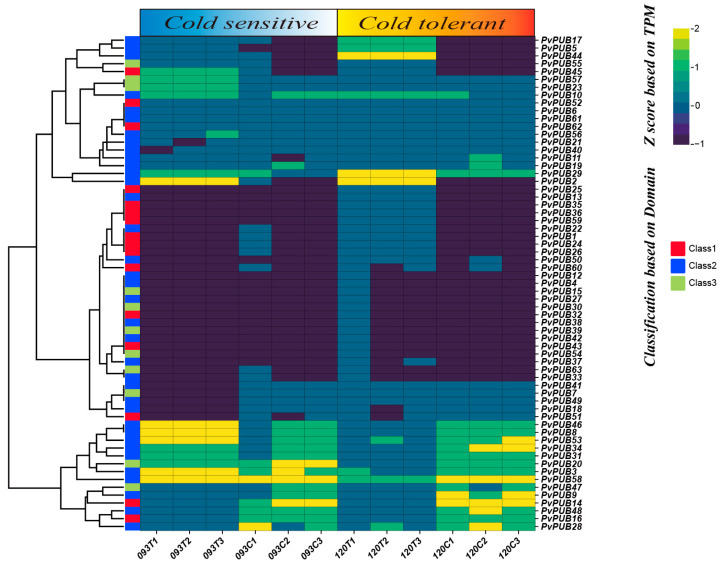
Heatmap of *PvPUB* gene expression patterns in cold-sensitive and cold-tolerant common bean varieties. The horizontal axis represents different samples, with T representing the treatment group (each with 3 replicates) and C representing the control group (each with 3 replicates). The vertical axis represents the *PvPUB* genes. Colors indicate gene expression Z-scores: yellow for high expression, green for moderate expression, and purple for low expression. The dendrogram on the left clusters genes based on their expression patterns. Color bars (red, blue, and green) indicate gene classification by domain, corresponding to Classes 1, 2, and 3, respectively. Data were normalized using TPM and visualized with Chiplot.

**Table 1 ijms-25-07968-t001:** Members of the *PvPUB* Gene Family in the common bean (*Phaseolus vulgaris* L.) Genome.

Genomic ID	Gene Name	Class	Isoelectric Point	Molecular Weight	Cellular Localization	Instability Index	Aliphatic Index	GRAVY	Protein Length (aa)	Gene Size	Chromosomal Position
Phvul.001G196000.1.v2.1	*PvPUB1*	Class 1	8.7	46,436.14	Cytoplasm	35.56	102.15	−0.038	418	1604	1
Phvul.001G228500.1.v2.1	*PvPUB2*	Class 2	8.26	49,453.27	Mitochondrion	34.74	99.6	−0.099	446	1660	1
Phvul.001G260300.1.v2.1	*PvPUB3*	Class 2	5.7	84,556.93	Nucleus	47.29	93.98	−0.171	766	4958	1
Phvul.002G031400.1.v2.1	*PvPUB4*	Class 2	6.32	85,204.42	Nucleus	48.49	95.53	−0.294	758	5400	2
Phvul.002G072000.1.v2.1	*PvPUB5*	Class 2	8.77	79,577.39	Chloroplast	46.14	100.62	−0.056	712	2138	2
Phvul.002G092600.1.v2.1	*PvPUB6*	Class 2	6.85	71,532.82	Nucleus	45.97	102.36	−0.124	656	3345	2
Phvul.002G146600.1.v2.1	*PvPUB7*	Class 3	6.92	92,027.45	Chloroplast	48.94	85.11	−0.506	803	4315	2
Phvul.002G175000.1.v2.1	*PvPUB8*	Class 2	5.22	84,772.85	Nucleus	52.2	89.7	−0.199	763	4688	2
Phvul.002G219800.1.v2.1	*PvPUB9*	Class 2	8.48	40,001.25	Plasma membrane	49.11	110.82	0.083	368	1693	2
Phvul.002G241800.1.v2.1	*PvPUB10*	Class 2	7.17	78,518.64	Chloroplast	47.7	97.99	0.036	711	3759	2
Phvul.002G270300.1.v2.1	*PvPUB11*	Class 2	7.99	46,301.58	Nucleus	49.56	106.03	0.017	418	1880	2
Phvul.002G311800.1.v2.1	*PvPUB12*	Class 2	5.49	148,371.85	Chloroplast	50.3	96.57	−0.14	1334	8342	2
Phvul.002G331800.1.v2.1	*PvPUB13*	Class 2	8.26	49,892.08	Chloroplast	41.96	106.24	−0.035	444	1334	2
Phvul.003G048200.1.v2.1	*PvPUB14*	Class 1	7.15	45,242.43	Cytoplasm	51.12	109.83	0.121	413	1241	3
Phvul.003G134500.1.v2.1	*PvPUB15*	Class 3	5.71	85,778.69	Chloroplast	40.3	91.73	−0.294	763	5302	3
Phvul.003G138252.1.v2.1	*PvPUB16*	Class 1	5.49	112,675.95	Chloroplast	41.07	110.43	−0.095	1013	4644	3
Phvul.003G199200.1.v2.1	*PvPUB17*	Class 2	8.48	75,024.85	Plasma membrane	39.69	104.81	0.019	682	2491	3
Phvul.003G233800.1.v2.1	*PvPUB18*	Class 2	6.12	48,522.88	Cytoplasm	41.53	117.65	0.249	443	1331	3
Phvul.003G254600.1.v2.1	*PvPUB19*	Class 2	6.27	84,773.76	Nucleus	42.56	95.32	−0.202	757	4089	3
Phvul.004G033100.1.v2.1	*PvPUB20*	Class 3	5.48	117,694.74	Nucleus	43.27	92.7	−0.223	1042	9823	4
Phvul.004G098600.1.v2.1	*PvPUB21*	Class 2	8.77	44,220.42	Chloroplast	44.05	103.27	0.107	407	2018	4
Phvul.004G147500.1.v2.1	*PvPUB22*	Class 2	6.64	49,159.99	Chloroplast	46.64	102.75	−0.116	437	1703	4
Phvul.005G008900.1.v2.1	*PvPUB23*	Class 3	6.72	89,958.55	Chloroplast	44.36	88.97	−0.316	808	8517	5
Phvul.005G119100.1.v2.1	*PvPUB24*	Class 1	8.96	45,111.03	Cytoplasm	47.53	113.23	0.137	403	1484	5
Phvul.005G119200.1.v2.1	*PvPUB25*	Class 1	8.79	46,597.19	Cytoplasm	48.56	102.69	−0.073	413	1389	5
Phvul.006G068700.2.v2.1	*PvPUB26*	Class 1	6.19	99,632.12	Nucleus	40.87	101.29	−0.129	901	6187	6
Phvul.006G076000.1.v2.1	*PvPUB27*	Class 2	5.45	90,667.38	Chloroplast	51.18	96.3	−0.248	838	9544	6
Phvul.006G107800.1.v2.1	*PvPUB28*	Class 2	7.19	49,097.58	Nucleus	39.85	105.51	−0.207	443	1968	6
Phvul.006G171900.1.v2.1	*PvPUB29*	Class 2	6.85	78,355.28	Chloroplast	45.26	98.98	0.014	716	3290	6
Phvul.007G072300.1.v2.1	*PvPUB30*	Class 3	5.8	83,383.05	Nucleus	38.33	92.87	−0.298	735	9461	7
Phvul.007G098700.1.v2.1	*PvPUB31*	Class 2	6.17	68,934.26	Cytoplasm	33.92	102.23	−0.205	631	3559	7
Phvul.007G115800.1.v2.1	*PvPUB32*	Class 1	6.23	50,705.33	Chloroplast	35.65	106.3	−0.107	457	1786	7
Phvul.007G136300.1.v2.1	*PvPUB33*	Class 2	5.43	167,076.4	Nucleus	41.38	90.79	−0.282	1491	7869	7
Phvul.007G192300.1.v2.1	*PvPUB34*	Class 2	7.25	87,205.42	Chloroplast	49.48	94.19	−0.292	794	4975	7
Phvul.007G267700.1.v2.1	*PvPUB35*	Class 1	8.64	45,041.84	Cytoplasm	39.69	113.19	0.118	411	1698	7
Phvul.007G267800.1.v2.1	*PvPUB36*	Class 1	8.73	45,836.99	Cytoplasm	34.52	107.83	0.033	405	2056	7
Phvul.008G002200.2.v2.1	*PvPUB37*	Class 2	5.64	81,145.87	Nucleus	42.42	94.94	−0.199	727	6199	8
Phvul.008G059800.1.v2.1	*PvPUB38*	Class 2	5.47	71,753.7	Golgi apparatus	51.11	100.74	−0.256	637	3036	8
Phvul.008G062200.1.v2.1	*PvPUB39*	Class 3	5.82	98,854	Cytoplasm	52.14	87.38	−0.419	883	7490	8
Phvul.008G105675.1.v2.1	*PvPUB40*	Class 2	5.69	70,209.67	Endoplasmic Reticulum	36.6	103.35	−0.05	639	15,298	8
Phvul.008G134400.1.v2.1	*PvPUB41*	Class 2	8.55	51,408.65	Cytoplasm	43.35	102.39	−0.234	460	3437	8
Phvul.008G134500.1.v2.1	*PvPUB42*	Class 2	5.21	33,113.88	Cytoplasm	39.73	89.37	−0.385	287	3755	8
Phvul.008G222100.1.v2.1	*PvPUB43*	Class 1	7.15	74,729.23	Nucleus	43.05	105.52	−0.059	683	2575	8
Phvul.008G238500.1.v2.1	*PvPUB44*	Class 2	8.59	48,350.49	Plasma membrane	38.02	100.76	0.022	437	2326	8
Phvul.008G249400.1.v2.1	*PvPUB45*	Class 1	5.74	83,718.27	Peroxisome	41.15	101.09	−0.186	769	7450	8
Phvul.009G012400.1.v2.1	*PvPUB46*	Class 2	5.15	89,127.63	Plasma membrane	41.36	110.1	0.043	814	5590	9
Phvul.009G043900.1.v2.1	*PvPUB47*	Class 3	4.72	46,563.18	Nucleus	56.78	60.4	−0.709	426	4903	9
Phvul.009G048000.1.v2.1	*PvPUB48*	Class 2	5.91	112,968.0	Golgi apparatus	47.48	111.79	−0.068	1005	6044	9
Phvul.009G073300.1.v2.1	*PvPUB49*	Class 2	6.94	46,648.8	Chloroplast	48.4	117.58	0.21	426	2138	9
Phvul.009G148200.1.v2.1	*PvPUB50*	Class 2	8.87	45,016.58	Chloroplast	42.47	111.02	0.15	412	2002	9
Phvul.009G156600.1.v2.1	*PvPUB51*	Class 1	7.53	43,758.33	Cytoplasm	41.98	110.94	0.152	392	1178	9
Phvul.009G181600.1.v2.1	*PvPUB52*	Class 1	8.01	58,455.76	Chloroplast	55.78	92.2	−0.237	533	5363	9
Phvul.010G029900.1.v2.1	*PvPUB53*	Class 2	5.71	31,658	Cytoplasm	36.64	85.72	−0.49	278	10,297	10
Phvul.010G099400.2.v2.1	*PvPUB54*	Class 3	6.73	99,946.7	Nucleus	49.48	82.61	−0.438	890	6083	10
Phvul.010G100901.1.v2.1	*PvPUB55*	Class 3	6.41	51,899.13	Nucleus	39.83	87.13	−0.361	460	2059	10
Phvul.010G116400.1.v2.1	*PvPUB56*	Class 2	5.65	111,409.7	Cytoplasm	46.67	108.79	−0.087	997	6901	10
Phvul.011G064400.1.v2.1	*PvPUB57*	Class 3	6.12	90,165.7	Nucleus	48.52	83.89	−0.3	800	7113	11
Phvul.011G064700.1.v2.1	*PvPUB58*	Class 2	5.44	72,823.32	Endoplasmic Reticulum	36.28	99.45	−0.217	670	4996	11
Phvul.011G099300.1.v2.1	*PvPUB59*	Class 1	9.31	46,252.98	Cytoplasm	33.11	100.59	−0.069	410	1232	11
Phvul.011G099400.1.v2.1	*PvPUB60*	Class 1	8.51	44,272.07	Cytoplasm	42.33	110.43	0.182	399	1505	11
Phvul.L001610.1.v2.1	*PvPUB61*	Class 2	7.02	74,759.11	Chloroplast	43.97	103.61	0.072	689	2069	scaffold_1097
Phvul.L001616.1.v2.1	*PvPUB62*	Class 1	7.22	61,447.63	Nucleus	57.38	91.74	−0.26	562	1688	scaffold_1123
Phvul.L005143.1.v2.1	*PvPUB63*	Class 3	6.42	91,771	Chloroplast	50.92	86.97	−0.27	819	4320	scaffold_15

## Data Availability

Data are contained within the article.

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
