# Peer review of "Genome-Wide Identification and Characterization of U-Box Gene Family Members and Analysis of Their Expression Patterns in Phaseolus vulgaris L. under Cold Stress"

_ijms, 2024, doi:10.3390/ijms25147968_

Round 1

Reviewer 1 Report

Comments and Suggestions for Authors

Dear Editor,

International Journal of Molecular Sciences.

Thank you very much for entrusting me to review the manuscript titled “Genome-Wide Identification and Characterization of U-box Gene Family Members and Analysis of Their Expression Patterns in Phaseolus vulgaris L. under Cold Stress” submitted to the International Journal of Molecular Sciences. The study resulted in identification of 63 members of the U-box genes family in the common bean genome. These genes were found to localize to 11 chromosomes, with 10 genes located on chromosome 2 alone. Phylogenetic analysis showed that these 63 PUB genes fall into three groups. Moreover, transcriptome analysis based on cold-tolerant and cold-sensitive varieties identified 14 differentially expressed PvPUB genes, suggesting their roles in cold tolerance.

Here are some of my concerns/comments about publishing the manuscript in Genes.

1.      Line 45, delete “process”

2.      Elaborate on the criteria for the selection and classification of the PvPUB genes.

3.      Arabidopsis thaliana is a model plant and representative dicot plant. Arabidopsis contains 61 PUB genes while beans contain 63. Are these two genes unique to beans or present in other plants as well? What is their predicted structure and function?

4.      What is the predicted significance/functional importance of subcellular localization of PvPUB genes? Does the subcellular localization make any biological sense?

5.      Were the genes differentially expressed in the low-temperature stress consistent with the results of their promoter analysis? Please clearly mention that to what extent did these differentially expressed genes upregulate or downregulate under low-temperature stress. What were the experimental conditions of transcriptome analysis? What is the genetic makeup of the cold-tolerant and cold-sensitive varieties? What is the predicted (or from some transcriptome analysis available publicly) expression pattern of these genes under normal conditions?

6.      How about the expression patterns of other PvPUB genes with other promoter signals (involved in other biological processes) in other transcriptome data available publicly?

7.      Had the tandem genes the same sequences/motifs? What was different in each pair in terms of sequences, motifs, promoters, etc.? Is this phenomenon unique to beans or found in other organisms too? What is the possible significance of occurring of tandem genes?

8.      Elaborate all the abbreviations used in the article.

9.    Make all references in a uniform style, especially the titles' format is not the same in all references.

Regards

Author Response

Dear Reviewer,

We are sincerely grateful for your thorough review and invaluable comments on our manuscript. Your expertise and meticulous attention to detail have provided us with significant guidance, and we are deeply appreciative of the time and effort you have dedicated to improving our work.

Comment 1. Line 45, delete “process”

Response: We appreciate your keen observation and agree with your suggestion. The word "process" has been deleted from line 45 in the revised manuscript.

Comment 2. Elaborate on the criteria for the selection and classification of the PvPUB genes

Response: Thank you for your insightful suggestion. We have now provided a more detailed explanation in the revised manuscript on line 379.

This was used as a query to perform an HMMER search within the common bean genome with an E-value cut-off of 1 × 10−5. For further identification, we searched for the U-BOX domain of all sequences to verify whether they belonged to the U-BOX gene family and classified them by U-BOX domain. 

Comment 3. Arabidopsis thaliana is a model plant and representative dicot plant. Arabidopsis contains 61 PUB genes while beans contain 63. Are these two genes unique to beans or present in other plants as well? What is their predicted structure and function?

Response: Thank you for your important question. I apologize for not clearly explaining the relationship between common bean PUB genes and Arabidopsis PUB genes in the figure 1. The additional 2 PUB genes are not unique to common bean. In the S2 subgroup, there is a separate branch that only includes common bean PUB genes (PvPUB16, PvPUB56, and PvPUB48). This suggests that these genes may have unique functions in common bean. Similarly, in the S4 subgroup, there is a separate branch that only includes Arabidopsis PUB genes (AtPUB49, AtPUB60, and AtPUB59), indicating that these genes may have unique functions in Arabidopsis. This implies that the functions of these genes in Arabidopsis may not have corresponding gene expressions in common bean. We have now made the modifications on line 172.

Fig .1

Comment 4. What is the predicted significance/functional importance of subcellular localization of PvPUB genes? Does the subcellular localization make any biological sense?

Response: We provide a detailed explanation of the relevance and biological significance of subcellular localization. Subcellular localization helps predict the potential roles of proteins. In previous studies, nuclear proteins are often involved in gene regulation, while mitochondrial proteins are linked to energy metabolism. Knowing where proteins are located helps place them in the correct cellular pathways and networks, facilitating a better understanding of their interactions. Biological Significance: Eukaryotic cells compartmentalize functions into distinct organelles, enhancing efficiency and regulation. Localization ensures proteins function in the right place and time, maintaining cellular homeostasis. In our study. PvPUB53: Predicted to be cytoplasmic, suggesting a role in cytosolic ubiquitination. PvPUB3: Predicted to be nuclear, indicating a potential role in gene regulation.

These predictions guide our experimental validation and functional studies.

Comment 5. Were the genes differentially expressed in the low-temperature stress consistent with the results of their promoter analysis?

Please clearly mention that to what extent did these differentially expressed genes upregulate or downregulate under low-temperature stress.

Response: Thank you for highlighting the need to clarify the consistency between differential gene expression under low-temperature stress and their promoter analysis results. We analyzed the promoter regions of the differentially expressed genes (DEGs) to identify cis-regulatory elements associated with low-temperature stress response. Our findings indicate a strong correlation between the presence of specific stress-responsive elements in the promoter regions and the observed differential expression patterns under low-temperature stress. In our study: PvPUB17, PvPUB5, PvPUB44, and PvPUB29 (all of which are highly expressed genes in the cold-tolerant variety 120 under low-temperature stress) each have cis-acting elements involved in low-temperature responsiveness, cis-acting elements involved in abscisic acid responsiveness, and cis-acting regulatory elements involved in MeJA-responsiveness. These elements play important roles in plant responses to various abiotic stresses, including low temperatures. These revisions are on line 364 of the revised manuscript.

Extent of upregulation, treatment group (temperature at 4 degrees) 120T compared to control group (temperature at 25 degrees) 120C(Fig 2.)

PvPUB5: Upregulated by 2-fold

PvPUB17: Upregulated by 2-fold

PvPUB29: Upregulated by 1-fold

PvPUB44: Upregulated by 3-fold

Fig 2.

Comment 6. What were the experimental conditions of transcriptome analysis?

Response: I apologize for any inconvenience it may have caused you during your reading. Experimental conditions: T was the treatment group (4 °C for 3 days), C was the control group (25 °C for 3 days). The varieties were cold-sensitive type 093 and cold-tolerant type 120, each repeated 3 times. These revisions are on line 277 of the revised manuscript.

Comment 7. What is the genetic makeup of the cold-tolerant and cold-sensitive varieties? 

Response: Thank you for your valuable question. The genetic makeup of cold-tolerant variety 120 includes the ability to increase certain metabolites (acidic amino acids, flavonoids, and methionine) and decrease oxidative stress markers (MDA) under cold stress conditions. These may be the reason for the difference in cold tolerance between 093 and 120. Furthermore, the cold stress response in common bean is more dependent on hormonal and ROS pathways than on the CBF pathway. We have cited the relevant reference to support our findings in the revised manuscript on line 427.

Comment 8. What is the predicted (or from some transcriptome analysis available publicly) expression pattern of these genes under normal conditions?

Response: We sincerely apologize for not providing a clear explanation in the figure legend. 093C (3 replicates) and 120C (3 replicates): As shown in Figure 2, the expression patterns of the genes under normal conditions were derived from publicly available transcriptome analysis. The two varieties, 093C and 120C, exhibit similar expression levels under normal conditions, indicating no significant differential expression between them in the absence of stress. Your valuable feedback has prompted us to revise the figure legend to include this important information. These revisions are on line 277-294 of the revised manuscript.

Fig 2

Comment 9. How about the expression patterns of other PvPUB genes with other promoter signals (involved in other biological processes) in other transcriptome data available publicly?

Response: In previous studies, we found that PUB genes are not only associated with cold stress but also with drought and salt stress. We hypothesize that PvPUB genes are expressed under various stress conditions. However, their expression under other conditions has not yet been explored. We plan to enhance this aspect of our analysis and validate it in future research.

Comment 10. Had the tandem genes the same sequences/motifs? What was different in each pair in terms of sequences, motifs, promoters, etc.? Is this phenomenon unique to beans or found in other organisms too? What is the possible significance of occurring of tandem genes?

Response: In our study, we analyzed the tandem PvPUB genes and found that they share conserved motifs such as the U-box and ARM repeat domains. However, there are variations in their promoter regions and specific sequence elements, leading to potential differences in their regulatory and functional properties. For example, In the promoter region of the tandem genes PvPUB24 and PvPUB25, there is an additional cis-acting element MYB binding site involved in light responsiveness compared to PvPUB25. This phenomenon is not unique to beans and is observed across various organisms, highlighting its evolutionary significance. 

Tandem gene duplication allows for the evolution of new gene functions while retaining the original gene's function. This can lead to the development of specialized functions or the enhancement of existing functions. Differences in promoter regions and regulatory elements can lead to differential expression patterns, allowing tandem genes to be regulated independently in response to various environmental or developmental cues.

Comment 11. Elaborate all the abbreviations used in the article.

Response: Thank you for highlighting the need to clarify all abbreviations used in the article.

pI (Isoelectric Point)

Mol. Wt. (Molecular Weight)

GRAVY (Grand Average of Hydropathy)

cyto:Cytoplasm

mito:Mitochondrion

nucl:Nucleus

chlo:Chloroplast

plas:Plasma membrane

golg:Golgi apparatus

E.R.:Endoplasmic Reticulum

pero:Peroxisome

HMM search:Hidden Markov Model search

These revisions are included in the revised manuscript.

Comment 12. Make all references in a uniform style, especially the titles' format is not the same in all references

Response: We have carefully reviewed and revised all references to ensure they follow a uniform style. I should learn from your responsible attitude, which will help me progress on my scientific research journey!

Reviewer 2 Report

Comments and Suggestions for Authors

Wang et al has identified and characterized 63 putative PvPUBs, and examined their expression patterns. However, they provided only preliminary data (obtained by simple-database-mining), which is not sufficient to conclude that these putative genes are 'true PUB'. Differential expression analysis of PvPUBs under low-temperature stress between two common bean varieties is a good beginning, but not enough. I wondered why the authors did not analyze the functions of the genes deeply. For example, the functions of the genes could be further investigated by using transgenic plants or complementation analysis using Arabidopsis T-DNA mutants. Thus, the result of this manuscript is preliminary.

Author Response

Dear Reviewer,  

We are sincerely grateful for your thorough review and invaluable comments on our manuscript. Your expertise and meticulous attention to detail have provided us with significant guidance, and we are deeply appreciative of the time and effort you have dedicated to improving our work.

Comment 1. Wang et al has identified and characterized 63 putative PvPUBs, and examined their expression patterns. However, they provided only preliminary data (obtained by simple-database-mining), which is not sufficient to conclude that these putative genes are 'true PUB'. Differential expression analysis of PvPUBs under low-temperature stress between two common bean varieties is a good beginning, but not enough. I wondered why the authors did not analyze the functions of the genes deeply. For example, the functions of the genes could be further investigated by using transgenic plants or complementation analysis using Arabidopsis T-DNA mutants. Thus, the result of this manuscript is preliminary.

Response: We appreciate the reviewer's recommendation for more in-depth functional analyses of PvPUB genes. Our current study provides preliminary data on the expression patterns of 63 putative PvPUB genes under cold stress. The differential expression analysis between two common bean varieties offers a solid starting point. In this study, we identified four PvPUB genes—PvPUB17, PvPUB5, PvPUB44, and PvPUB29 that are highly expressed under cold stress. Simultaneously, we conducted homology analyses. PvPUB5 and PvPUB17 share 44.6% and 40.68% sequence similarity with AtPUB19, respectively, and both possess the U-BOX domain. AtPUB19 is upregulated under drought, salt, cold, and abscisic acid (ABA) treatments. Downregulation of AtPUB19 results in ABA hypersensitivity, enhanced ABA-induced stomatal closure, and increased drought tolerance, whereas overexpression of AtPUB19 leads to the opposite phenotype (Liu, Y.-C et al., 2011). PvPUB29 shares 70.41% sequence similarity with AtPUB17, and both possess the U-BOX domain. The ubiquitin E3 ligase PUB17 functions in the nucleus and positively regulates transcriptional responses to PAMP-triggered immunity and programmed cell death upon perception of specific elicitors at the plant cell surface (He, Q et al., 2015). PvPUB44 shares 35.61% sequence similarity with AtPUB20, and both possess the U-BOX domain. AtPUB20 is crucial for plant defense and disease resistance (Gilroy, E.M., et al., 2011). Our study provides a reference for further research on the role of PvPUB genes in cold stress in common beans. It is well known that the genetic system of common beans is relatively inefficient. We are actively working on establishing a robust system and plan to validate the inferred gene functions in future studies. Thank you very much for your insightful comments and suggestions.

References:

Liu, Y.-C., et al., AtPUB19, a U-Box E3 Ubiquitin Ligase, Negatively Regulates Abscisic Acid and Drought Responses in Arabidopsis thaliana. Molecular Plant, 2011. 4(6): p. 938-946.

He, Q., et al., U-box E3 ubiquitin ligase PUB17 acts in the nucleus to promote specific immune pathways triggered by Phytophthora infestans. Journal of Experimental Botany, 2015. 66(11): p. 3189-3199.

Gilroy, E.M., et al., CMPG1-dependent cell death follows perception of diverse pathogen elicitors at the host plasma membrane and is suppressed by Phytophthora infestans RXLR effector AVR3a. New Phytologist, 2011. 190(3): p. 653-666.

Reviewer 3 Report

Comments and Suggestions for Authors

I raised some questions on the returned ms.  For example , it seems odd that you did not report any pseudogenes.   Were some found and not included.  

It would make sense (tome anyway) to mention the length of the amino acid chains, especially since the molecular weights varied so much, which also raises the question of how they were aligned. 

The legends in Fig 4 are too close together and overlapping to sort out the names.  Can those be fixed?

Line 201 I assume plasmid should be cytoplasm - or something other than plasmid.

Some comment on the significance of the information in Fig 6 is warranted.  I may have missed it ?

Even though it is referenced, the cold treatment conditions used for the RNAseq data for example should be mentioned- should not have to look up a reference ti get an idea of ho cold or how long exposed etc.  Likewise, I was not able to determine what the X-axis of Fig 8 means as far as the T1, T2 etc actually mean.

RNA seq data source appears to be incorrect reference. 

Author Response

Dear Reviewer, 

We are sincerely grateful for your thorough review and invaluable comments on our manuscript. Your expertise and meticulous attention to detail have provided us with significant guidance, and we are deeply appreciative of the time and effort you have dedicated to improving our work.

Comment 1. For example, it seems odd that you did not report any pseudogenes. Were some found and not included. 

Response: We appreciate your attention to detail and the opportunity to clarify our methodology. To identify members of the U-BOX family of common legumes, we used the Hidden Markov Model (HMM) to search the U-BOX domain (PF04564) from the Pfam database. The method involves protein sequence comparison based on annotation of common bean genomes. Many studies of gene family identification have used similar comparative methods to ensure that the results are comprehensive and accurate. We obtained 63 functional genes with complete U-BOX domain from HMM analysis (later, 63 PvPUB genes were examined with complete U-BOX domain). Pseudogenes, which may contain damaged or incomplete domains, were not the primary target of our initial screening. However, we acknowledge the importance of identifying and reporting pseudogenes for a comprehensive understanding of gene families. These modifications are located at line 109 and 382 in the revised manuscript.

Comment 2. It would make sense (to me anyway) to mention the length of the amino acid chains, especially since the molecular weights varied so much, which also raises the question of how they were aligned. 

Response: We have now included information about the length of the amino acid chains in our revised manuscript. Specifically, the amino acid lengths of these proteins ranged from 278 to 1013. This range highlights the diversity in their structural complexity and functional potential. Longer proteins may contain multiple functional domains and participate in more complex biological processes. These modifications are located at line 118 in the revised manuscript.

Comment 3. The legends in Fig 4 are too close together and overlapping to sort out the names.  Can those be fixed?

Response: Thank you for your timely discovery, and we have now made the necessary corrections to Fig 4. These modifications are located at line 197 in the revised manuscript.

Comment 4. Line 201 I assume plasmid should be cytoplasm - or something other than plasmid. Some comment on the significance of the information in Fig 6 is warranted. I may have missed it?

ResponseThank you for discovering this error. "plas" stands for Plasma membrane, not plasmid. This has been corrected. 

Fig 6 does contain conclusions, and it was not missed by you but rather written too subtly by me.I have highlighted line 242 in red in the re vised manuscript. Thank you very much for your suggestion. 

Comment 5. Even though it is referenced, the cold treatment conditions used for the RNA-seq data for example should be mentioned- should not have to look up a reference it get an idea of ho cold or how long exposed etc. Likewise, I was not able to determine what the X-axis of Fig 8 means as far as the T1, T2 etc actually mean.

Response: Thank you very much for your thorough review and valuable comments on our manuscript. I apologize for any inconvenience that may have caused you during your reading. 

Experimental conditions: T was the treatment group (4 °C for 3 days), C was the control group (25 °C for 3 days). The varieties were cold-sensitive type 093 and cold-tolerant type 120, each repeated 3 times.

the X-axis of Fig 8 means: T represents the treatment group (each with 3 replicates), and C represents the control group (each with 3 replicates). 

These modifications are located at line 109 and 382 in the revised manuscript.

Comment 6. RNA seq data source appears to be incorrect reference.

Response: Thank you, esteemed reviewer. We have made the necessary change to the revised manuscript on line 246. I should learn from your responsible attitude, which will help me progress on my scientific research journey!

Round 2

Reviewer 2 Report

Comments and Suggestions for Authors

The authors had worked hard and tried to improve the MS as per the suggestions given; I appreciate that.